# Chemical Composition and Variability of the Volatile Components of *Myrciaria* Species Growing in the Amazon Region

**DOI:** 10.3390/molecules27072234

**Published:** 2022-03-30

**Authors:** Jamile Silva da Costa, Waldemir Magno S. Andrade, Raphael O. de Figueiredo, Paulo Vinicius L. Santos, Jofre Jacob da Silva Freitas, William N. Setzer, Joyce Kelly R. da Silva, José Guilherme S. Maia, Pablo Luis B. Figueiredo

**Affiliations:** 1Programa de Pós-Graduação em Ciências Farmacêuticas, Universidade Federal do Pará, Belém 66075-900, Brazil; jamile.costa@ics.ufpa.br (J.S.d.C.); gmaia@ufpa.br (J.G.S.M.); 2Universidade do Estado do Pará Campus Cametá, Cametá 68400-000, Brazil; waldemir.andrade@aluno.uepa.br; 3Laboratório de Química dos Produtos Naturais, Centro de Ciência Biológicas e da Saúde, Universidade do Estado do Pará, Belém 66087-662, Brazil; raphael.figueiredo@aluno.uepa.br (R.O.d.F.); paulo.ssantos@aluno.uepa.br (P.V.L.S.); 4Laboratório de Morfofisiologia Aplicada a Saúde, Departamento de Morfologia e Ciências Fisiológicas, Universidade do Estado do Pará, Belém 66087-662, Brazil; jofre.freitas@uepa.br; 5Department of Chemistry, University of Alabama in Huntsville, Huntsville, AL 35899, USA; wsetzer@chemistry.uah.edu; 6Aromatic Plant Research Center, 230 N 1200 E, Suite 100, Lehi, UT 84043, USA; joycekellys@ufpa.br; 7Programa de Pós-Graduação em Biotecnologia, Universidade Federal do Pará, Belém 66075-900, Brazil; 8Programa de Pós-Graduação em Química, Universidade Federal do Maranhão, São Luís 64080-040, Brazil; 9Departamento de Ciências Naturais, Centro de Ciência Sociais e Educação, Universidade do Estado do Pará, Belém 66050-540, Brazil

**Keywords:** essential oils, mono- and sesquiterpenes, chemotaxonomy, multivariate analyses

## Abstract

*Myrciaria* (Myrtaceae) species have been well investigated due to their chemical and biological relevance. The present work aimed to carry out the chemotaxonomic study of essential oils of the species *M. dubia*, *M. floribunda*, and *M. tenella*, sampled in the Brazilian Amazon and compare them with the volatile compositions from other *Myrciaria* species reported to Brazil and Colombia. The leaves of six *Myrciaria* specimens were collected (PA, Brazil) during the dry season, and their chemical compositions were analyzed by gas chromatography-mass spectrometer (GC-MS) and gas chromatography-flame ionization detector (GC-FID). The main compounds identified in the essential oils were monoterpenes with pinane and menthane skeletons, followed by sesquiterpenes with caryophyllane and cadinane skeletons. Among the sampled *Myrciaria* specimens, five chemical profiles were reported for the first time: profile I (*M. dubia*, α-pinene, 54.0–67.2%); profile II (*M. floribunda*, terpinolene 23.1%, α-phellandrene 17.7%, and γ-terpinene 8.7%); profile III (*M. floribunda*, γ-cadinene 17.5%, and an unidentified oxygenated sesquiterpene 15.0%); profile IV (*M. tenella*, *E*-caryophyllene 43.2%, and α-humulene 5.3%); and profile V (*M. tenella*, *E*-caryophyllene 19.1%, and caryophyllene oxide 41.1%). The *Myrciaria* chemical profiles showed significant variability in extraction methods, collection sites, plant parts, and genetic aspects.

## 1. Introduction

*Myrciaria* comprises 31 species growing in Argentina, Paraguay, Peru, Bolivia, Brazil, and Australia [1], belonging to the Myrtaceae, which has a significant botanic representation, with 142 genera and about 3500 tropical and subtropical species [2].

The species of Myrtaceae stand out for their economic potential for wood exploitation, a source of ornamental plants, and edible fruits [3]. In addition, the Myrtaceae species are essential oil-producers which show a chemical variability of cyclic mono- and sesquiterpenes, composed by *p*-menthane, pinane, bisabolane, germacrane, caryophyllane, cadinane, and aromadendrane skeletons in *Psidium* genus [4], and caryophyllane, germacrane, and pinane in *Eugenia* and *Syzygium* genera [5]. Myrtaceae essential oils have many applications in the pharmaceutical, food, and cosmetic industries because of their antifungal, antibacterial, cytotoxic, and insecticidal properties [6].

Essential oils are obtained by various methods, such as hydrodistillation, steam distillation, cold-pressing (in the case of *Citrus*), and microwave-assisted distillation [7]. The essential oil yield is usually meager quantitatively [8]. Moreover, other methods can be applied to extract volatile compounds, such as simultaneous distillation-extraction (SDE) [9] and the solid-phase microextraction method (SPME and headspace) [10].

Due to the chemical and biological relevance of Myrtaceae species, some reviews of volatile compounds and essential oils of some genera of Myrtaceae have been published in recent years. Among them are *Eugenia*, *Syzygium* [5], and *Psidium* [4]. In addition, another review has addressed one hundred volatile samples of Myrtaceae species, including four samples of *Myrciaria* [11]. Thus, the scarcity of studies related to the *Myrciaria* species and the growing interest in this genus is evidenced by the number of recent works. This work aimed to carry out the chemotaxonomic study of the species *Myrciaria dubia*, *M. floribunda*, and *M. tenella*, sampled in the Brazilian Amazon, based on the composition of its essential oils and the analysis of other volatile compounds described in the literature.

## 2. Results and Discussion

### 2.1. Chemical Variability in the Sampled Specimens

Among the six *Myrciaria* leaf samples collected, M. floribunda showed the highest essential oil yield (Mflo-1: 1.4%; Mflo-2: 1.5%), followed by *M. tenella* (Mten-1: 2.1%; Mten-2: 1.2%) and *M. dubia* (Mdub-1: 0.5%; Mdub-2: 1.1%), and GC-FID and GC-MS were used to identify their volatile constituents. One hundred and eleven components were identified in all essential oil samples, representing an average of 87% of the composition of the total oils (Table 1).

Monoterpene hydrocarbons were predominant in M. dubia (60.2–74.5%) essential oil. Oxygenated sesquiterpenes were the main constituents of M. floribunda (0–36.0%) and M. tenella (0–55.9%) samples. However, a M. tenella sample (Mten-1) predominated with sesquiterpenes hydrocarbons (60.8%).

The main compounds (>5%) identified in the essential oils were the monoterpene hydrocarbons with pinane (α-pinene, 0–67.2%) and menthane skeletons (terpinolene, 0–23.1%; α-phellandrene, 0–17.7%; γ-terpinene, 0–8.7%; p-cymene, 0–7.2%; and β-phellandrene, 0–6.6%), followed by the sesquiterpenes with caryophyllane (*E*-caryophyllene, 0.3–43.2%; caryophyllene oxide, 0.4–41.1%; and α-humulene, 0–5.3%), and cadinane skeletons (γ-cadinene, 0–17.5%), as depicted in Figure 1.

#### 2.1.1. *Myrciaria dubia*

*Myrciaria dubia* (Kunth) McVaugh, popularly known as camu-camu, is native to the Brazilian and Peruvian Amazon regions. This species’ leaves and fruit peels are used in Brazilian traditional medicine to treat diarrhea, female diseases, and labyrinthitis [14], while in Peru, *M. dubia* leaves are used to treat colds and arthritis [15]. Moreover, *M. dubia* fruits are considered a natural source of antioxidants by their significant content of ascorbic acid and phenolic compounds. Thus, this species has socioeconomic and nutritional potential [1,16].

Mdub-1 and Mdub-2 (Table 1) samples of *M. dubia* in this work, and those reported in the literature, Mdub-3 to Mdub-10 (Table A1), were classified in seven distinct chemical profiles according to the composition of their essential oils. The first profile (Mdub-1 and Mdub-2) was characterized by the highest amount of α-pinene (54.4–67.2%), from hydrodistilled specimens growing in Pará state, Brazil. Profile II was characterized by limonene (74.3%) and α-pinene (10.8%), which comprises the leaves’ volatile concentrate of a specimen (Mdub-6) collected in Caquetá, Colombia, obtained by simultaneous distillation–extraction (SDE) [17]. Profile III grouped fruits’ volatile concentrates (Mdub-3 and Mdub-5), obtained by liquid–liquid extraction (LLE) and solid-phase microextraction (HS-SPME), rich in limonene (23.9–40.8%) and *E*-caryophyllene (9.6–15.9%), from specimens collected in Caquetá, Colombia [18]. Profile IV includes samples of unripe and ripe fruits (Mdub-8 and Mdub-9) from a specimen collected in Amazonas State, Brazil, extracted by solid-phase microextraction (HS-SPME) and characterized by limonene (32.1–27.5%) and tricyclene (23.7–28.3%) [19]. Profile V was characterized by limonene (74.3%) and α-pinene (10.8%) from a leaves’ volatile concentrate, extracted by SDE from a specimen sampled in Caquetá, Colombia [19]. Profile VI was rich in α-pinene (66.2%) and limonene (23.7%), from fruits’ volatile concentrate (Mdub-10) of a specimen collected in Manaus, Brazil, extracted by HS-SPME [20]. Profile VII (Mdub-4) was characterized by limonene (32.2%) and α-terpineol (22.2%), from fruits’ volatile concentrate of a specimen sampled in Caquetá, Colombia, extracted by SDE [18]. It should be taken into account that the chemical variability of *M. dubia* specimens is related to different extraction methods, plant parts, and/or collection sites.

#### 2.1.2. *Myrciaria floribunda*

*Myrciaria floribunda* (H. West ex Willd.) O. Berg is known elsewhere as rumberry, camboim, and cambuí, growing naturally in the Central and South American continents [21,22]. Their fruits contain rutin, phenolic acids, and β-cryptoxanthin (pro-vitamin A) and are consumed in nature as jelly and in distilled beverages [23].

Mflo-1 and Mflo-2 samples (Table 1) reported in this work, and those from the literature pointed out as Mflo-3 to Mflo-11, were grouped in ten chemical profiles according to the composition of their essential oils. Profile I, composed of Mflo-1 oil, was characterized by the monoterpene hydrocarbons terpinolene (23.1%), α-phellandrene (17.7%), and γ-terpinene (8.7%). Profile II was associated with Mflo-2 oil, where it predominated the sesquiterpene hydrocarbon γ-cadinene (17.5%) and an unidentified oxygenated sesquiterpene (15.0%, MW 218). Profile III grouped the hydrodistilled oils Mflo-7 (stems) and Mflo-11 (flowers) from a specimen sampled in Rio de Janeiro, Brazil, characterized by significant contents of 2*E*,6*Z*-farnesol (13.1–16.1%), and 2*E*,6*E*-farnesyl acetate (19.9–13.4%) [24,25]. The other hydrodistilled oils, Mflo-3, Mflo-5, Mflo-6, and Mflo-9 (leaves) [26], Mflo-4 (fruits) [27], Mflo-8 (flowers) [25], and Mflo-10 (stems) [25], sampled in Rio de Janeiro, Brazil, were distinguished from one another, representing chemical profiles with the following characteristics: Profile IV (Mflo-3), *E*-nerolidol (32.4%) and β-selinene (9.8%); Profile V (Mflo-4), δ-cadinene (26.9%) and γ-cadinene (15.7%); Profile VI (Mflo-5), 1,8-cineole (10.4%) and β-selinene (8.4%); Profile VII (Mflo-6), 1,8-cineole (38.4%), γ-himachalene (7.0%); Profile VIII (Mflo-8), 1,8-cineole (22.8%), 2*E*,6*Z*-farnesol (16.1%); Profile IX (Mflo-9), γ-himachalene (7.0%), α-terpineol (5.5%); Profile X (Mflo-10), germacra-4(15),5,10(14)-trien-1α-ol (19.9%) and 2*E*,6*E*-farnesyl acetate (13.1%). As was seen, the eleven *M. floribunda* specimens showed chemical variability in their essential oils, which can be related to the collection sites and parts of the extracted plants.

#### 2.1.3. *Myrciaria tenella*

*Myrciaria tenella* (DC.) O. Berg is a Brazilian native species known as cambuí, murta-do-campo and vassourinha [28,29]. Traditional communities have used their leaves in teas and postpartum uterine baths [29]. Moreover, their astringent and flavorful fruits are consumed in nature and juices or jellies [28,30].

The sample oils of *M. tenella* of this work (Mten-1 and -2, Table 1) presented sesquiterpenes with caryophyllane skeletons, such as *E*-caryophyllene (Mten-1, 43.2%; Mten-2, 19.1%), α-humulene (Mten-1, 5.3%; Mten-2, 2.3%), and caryophyllene oxide (Mten-1, 4.4%; Mten-2, 41.1%), as the main constituents. Seven chemical profiles were proposed for Mten-1 and Mten-2 oils samples, and those of literature, Mten-3 to Mten-7, were all from hydrodistilled leaves.

Profile I, corresponding to Mten-1 oil, was characterized by *E*-caryophyllene (43.2%) and α-humulene (5.3%), a sample collected in Baião, Pará, Brazil. Profile II was associated with Mten-2 oil, showing significant contents of caryophyllene oxide (41.1%) and *E*-caryophyllene (19.1%), a sample collected in Abaetetuba, Pará, Brazil. Profile III was composed of the Mten-3 oil, rich in *E*-caryophyllene (32.0%), 1,8-cineole (5.4%), and δ-cadinene (5.1%), sampled in Acará, Pará, Brazil [31]. The Mten-4 oil, from a sample collected in Mogi-Guaçu, São Paulo, Brazil, formed profile I, characterized by the sesquiterpenes *E*-caryophyllene (25.1%), spathulenol (9.7%), globulol (5.9%), and α-cadinol (5.2%) [32]. Profile V comprised the Mten-5 oil and was characterized by *E*-caryophyllene (11.4%), muurola-4,10(14)-dien-1β-ol (9.4%), and caryophyllene oxide (9.3%), a sample collected in Maracanã, Pará state, Brazil [33]. Profile VI, from Mten-6 oil, was characterized by α-pinene (25.1%), β-pinene (20.9%), *E*-caryophyllene (10.0%), and platiphyllol (8.9%), sampled in Rio de Janeiro, Brazil [34]. Profile VII, from Mten-7 oil, showed β-pinene (45.7%) as the primary constituents, a sample collected in Valinhos, São Paulo, Brazil [34].

As the specimen samples (Mten-1 to Mten-7) were all obtained by the same extraction method (hydrodistillation), it does not seem to have been the factor that influenced the variability of the chemical profiles and may be associated with the genetic aspects of their different locations of collection.

#### 2.1.4. *Myrciaria plinioides*

Two chemical profiles for *Myrciaria plinioides* D. Legrand essential oils (Mpli-1 and Mpli-2), rich in sesquiterpenes, were identified in the literature. Profile I (Mpli-1) was characterized by spathulenol (27.3%), α-copaene (9.5%), α-cadinol (8.6%), viridiflorol (8.5%), humulene epoxide II (7.2%), and cubenol (6.5%) [35]. In profile II, predominated spathulenol (21.1%), caryophyllene oxide (15.2%), isolongifolan-7-α-ol (9.8%), mustakone (5.6%), and α-cadinol (5.4%) [36]. Both leaf samples were collected in the Rio Grande do Sul, Brazil, and hydrodistilled to obtain their essential oils [35,36]. In the chemical variability of *M. plinioides*, it seems that a significant contribution of genetic factors and/or seasonality between the two samples can be inferred.

#### 2.1.5. *Myrciaria pilosa*

A single chemical profile (Mpil-1) was identified in the literature for *Myrciaria pilosa*. Sobral and Couto, characterized by guaiol (13.7%), *E*-caryophyllene (11.3%), and β-eudesmol (9.2%), whose essential oil was obtained by hydrodistillation from leaves of a specimen collected in Pernambuco, Brazil [37]. It was impossible to verify the major constituents’ chemical variability since the species is still poorly studied.

### 2.2. Multivariate Analyses of Myrciaria Species

The chemical variability of *Myrciaria* volatile samples was evaluated by multivariate statistical analyses (PCA, principal components analysis; HCA, hierarchical clusters analysis). The total percentage of monoterpene hydrocarbons (MH), oxygenated monoterpenes (OM), sesquiterpene hydrocarbons (SH), oxygenated sesquiterpenes (OS), and other compounds (OT) were obtained from oil samples, according to the original citations (Table 1 and Table A1). The data were used as variables (see Appendix B).

The HCA (Figure 2) shows the formation of two groups. The first one comprises two *Myrciaria plinioides* samples, ten *M. floribunda* samples, five *M. tenella* samples, and one *M. pilosa* sample. The second group was composed of all *M. dubia* samples, two *M. tenella* samples, and one *M. floribunda* sample.

The PCA total analysis (Figure 3) explained 66.6% of the data variability. The PC1 explained 40.8% of the data, showing positive correlations with monoterpene hydrocarbons (MH, λ = 0.661), oxygenated monoterpenes (OM, λ = 0.008) and other compounds (OT, λ = 0.280), and negative correlations with sesquiterpene hydrocarbons SH, (λ = −0.470) and oxygenated sesquiterpenes (OS, λ = −0.513). The PC2 explained 25.8% of the data, showing a positive correlation with sesquiterpene hydrocarbons (SH, λ =0.494) and other compounds (OT, λ = 0.509), and a negative correlation with monoterpene hydrocarbons (MH, λ = −0.033), oxygenated monoterpenes (OM, λ = −0.667) and oxygenated sesquiterpenes (OS, λ = −0.227). Similar to HCA, the PCA analysis confirmed the formation of two distinct groups.

Group I was characterized by the highest amounts of sesquiterpene hydrocarbons (7.7–92.1%) and oxygenated sesquiterpenes (4.6–88.2%), and minor amounts of monoterpene hydrocarbons (0–13.0%) and oxygenated monoterpenes (0–48.6%). Group II was characterized by the highest amounts of monoterpene hydrocarbons (39.6–93.8%), and minor amounts of oxygenated monoterpenes (0–27.1%) and sesquiterpene hydrocarbons (0.5–34.5%).

Group I was composed of *M. floribunda* oil samples (Mflo-2 to Mflo-11), excluding Mflo-1. This oil sample was the only one that displayed a significant amount of monoterpene hydrocarbons (78.2%). Therefore, a new chemotype is being described in this work for the first time. Likewise, *M. tenella* (Mten-1 to Mten-5) oil samples were associated with group I, excluding the Mten-6 and Mten-7 oils. These two oil samples were characterized by significant amounts of monoterpene hydrocarbons (Mten-6, 47.9%; Mten-7, 53.5%). Moreover, the two samples of *Myrciaria plinioides* (Mpli-1 e Mpli-2) were also included in group I by displaying low chemical variability, characterized by the presence of oxygenated sesquiterpenes (Mpli-1, 88.2%; Mpli-2, 24.1%).

Group II was composed of all oil samples of *Myrciaria dubia* (Mdub-1 to Mdub-10) and some oils of *M. tenella* (Mten-6 and Mten-7) and *M. floribunda* (Mflo-1), characterized by significant amounts of monoterpene hydrocarbons (39.6–93.8%).

### 2.3. Biological Activities of Myrciaria Species

Some studies have reported the biological properties of *Myrciaria* essential oils. The *M. tenella* leaf oil, rich in *E*-caryophyllene (25.1%), spathulenol (9.7%), globulol (5.9%), and α-cadinol (5.2%), showed an anti-inflammatory activity on chemotaxis assay, at the doses of 9 mg/mL, with 93% neutrophils inhibition [32]. Moreover, the *M. plinioides* oil, with significant contents of spathulenol (21.1%), caryophyllene oxide (15.2%), isolongifolan-7-α-ol (9.8%), mustakone (5.6%), and α-cadinol (5.4%), exhibited antileishmanial activity against promastigote forms of *Leishmania amazonensis* (IC_50_ 14.16 μg/mL) and *Leishmania infantum* (IC_50_ 101.50 μg/mL) [36].

The oil of *M. pilosa*, characterized by guaiol (13.7%), *E*-caryophyllene (11.3%), β-eudesmol (9.2%), and γ-eudesmol (6.6%), presented bactericidal properties against *Staphylococcus aureus* (MIC, 5 μg/mL) by the broth microdilution method, and antivirulence with reductions of 92.0% and 47.2%, respectively, in the hemolytic action and production of staphyloxanthin [37].

The *M. floribunda* oil showed potential to treat neurodegenerative diseases, acting as a modulator in the neurological system [27]. The floral oil, with 1,8-cineole (22.8%), 2*E*,6*Z*-farnesol (16.1%), 2*E*,6*E*-farnesyl acetate (13.4%), and linalool (12.7%) as the main constituents, and the leaf oil, rich in 1,8-cineole (38.4%), γ-himachalene (7.0%) and α-terpineol (5.5%), from an *M. floribunda* specimen collected on the coast of southeastern Brazil, showed an anti-acetylcholinesterase potential, with IC_50_ 1583 and 681 μg/mL, respectively [24]. In addition, the fruit peel oil of an *M. floribunda* specimen from northeastern Brazil, rich in δ-cadinene (26.9%), γ-cadinene (15.7%), γ-muurolene (6.2%), α-selinene (6.1%), α-muurolene (6.1%) and *E*-caryophyllene (5.5%), showed promise as an acetylcholinesterase inhibitor agent, with IC_50_ of 0.08 μg/mL [27]. Moreover, another oil of *M. floribunda*, characterized by spathulenol (21.1%), caryophyllene oxide (15.2%), isolongifolan-7-α-ol (9.8%), mustakone (5.6%) and α-cadinol (5.4%), displayed insecticidal activity against *Rhodnius prolixus*, the vector of Chagas disease, with LD_50_ between 19.51 and 742.49 μg/insect, from the 1st to the 30th days after the treatment [38].

### 2.4. Bibliometric Network Data

The bibliometric network map represents data of scientific bases, identifying the degree of connection among the various elements through the distance between their nodes. The smaller the distance, the greater the degree of connection. In addition, the node’s size indicates its relevance in the analyzed universe [39]. The co-occurrence of similar terms in titles, abstracts, and keywords of nineteen articles in the Scopus database from 2010 to 2021 were analyzed to relate and identify the most widespread themes about *Myrciaria* essential oils.

Figure 4 exhibits the generated map and its associations. The terms “essential oils”, “Myrtaceae”, “*Myrciaria floribunda*”, and “*Myrciaria tenella*” were the most frequent. Moreover, terms were grouped into five clusters. The largest cluster is represented in red and includes terms related to biological assays, such as “animal cell”, “mice” and “cytotoxicity”, and terms referring to chemical composition analysis, as “chemical analysis”, “mass fragmentography”, “α-cadinol” and “copaene”.

The second-largest cluster, in green, includes terms related to chemical composition analysis instrumentation, such as “gas chromatography” and “mass spectrometry”; and related terms to the chemical composition of essential oils, such as “essential oil composition”, “volatile organic compound”, “hydrocarbons”, “α-pinene”.

The third-largest cluster, in blue, has terms related to the chemical classes, such as “sesquiterpene” and “monoterpenes”, and terms referring to the insecticide activity, such as “insecticide” and “insect growth regulator”. The yellow cluster encompasses terms related to the extraction of essential oils, such as “distillation” and “hydrodistillation”. Finally, in purple, the fifth cluster shows terms related to the topic of this article, such as “essential oils” and “chemical variability”.

## 3. Material and Methods

### 3.1. Plant Material

The leaves of the six *Myrciaria* specimens were collected in Pará state, Brazil, month-by-month, during the dry season (August–December). The collection site, herbarium voucher number, and geographic coordinates are listed in Table 2. After identification, the plant specimens were deposited in the Herbarium of Museu Paraense Emílio Goeldi (MG) in the city of Belém, Brazil. The leaves were dried for three days at room temperature, ground, and then submitted to essential oil hydrodistillation in duplicate using a Clevenger-type apparatus. The oils obtained were dried over anhydrous sodium sulfate, and total oil yields were expressed as mL/100 g of the dried material [40,41].

### 3.2. Analysis of Essential Oil Composition

The oil composition analysis was performed by GC-MS, using a Shimadzu instrument Model QP-2010 ultra (Shimadzu, Tokyo, Japan), equipped with a Rtx-5MS (30 m × 0.25 mm; 0.25 μm film thickness) fused silica capillary column (Restek, Bellefonte, PA, USA). Helium was used as carrier gas adjusted to 1.0 mL/min at 57.5 KPa; split injection (split ratio 1:20) of 1 μL of *n*-hexane solution (oil 5 μL: *n*-hexane 500 μL); injector and interface temperature were 250 °C; oven programmed temperature was 60 to 240 °C (3 °C/min), followed by an isotherm of 10 min. EIMS (electron impact mass spectrometry): electron energy, 70 eV; ion source temperature was 200 °C. The mass spectra were obtained by automatically scanning every 0.3 s, with mass fragments in the range of 35–400 *m*/*z*. The compounds present in the samples were identified by comparison of their mass spectrum and retention index calculated for all volatile components using a linear equation by Van den Dool and Kratz [42], with the data present in the commercial libraries FFNSC-2 [13] and Adams [12]. The retention index was calculated using *n*-alkane standard solutions (C8–C40, Sigma-Aldrich, St. Louis, MO, USA) under the same chromatographic conditions. The GC-FID analysis was carried out on a Shimadzu QP-2010 instrument, equipped with an FID detector, in the same conditions, except that hydrogen was used as the carrier gas. The percentage composition of the oil samples was computed from the GC-FID peak areas. The analyses were carried out in triplicate.

### 3.3. Bibliographic Research Criteria

Bibliographic research was performed using Google Scholar, PubMed, Science Direct, Medline, and Scopus. Applied keywords were “*Myrciaria*”, “essential oil” and “volatile compound”. Some unusual or incorrect botanical names were updated based on “The Plant List” (http://www.theplantlist.org, accessed on 20 November 2021).

Bibliometric data analysis was done using more keywords to search for articles on the theme proposed in this review, using the VOSviewer software (version 1.6.15) [43]. The articles were downloaded from the databases in a supported format by the software. The primary data retrieved from the databases included information related to the article title, authors’ names, keywords, and citation information, including the reference lists. In this way, a cluster was generated relating the main keywords and their links with others used less frequently in the searches [4].

### 3.4. Multivariate Statistical Analyses

The multivariate statistical analysis was carried out to discern any relationship among *Myrciaria* oil samples (described in Appendix A). The total percentage of the compound classes monoterpene hydrocarbons (MH), oxygenated monoterpenes (OM), sesquiterpene hydrocarbons (SH), and oxygenated sesquiterpenes (OS), to each oil, was extracted from the original citations (Table A1). The data were used as variables (see Appendix B). The data matrix was standardized for the multivariate analysis by subtracting the mean and then dividing it by the standard deviation. Principal component analysis (PCA) was applied to verify the interrelation (free 390 version, Minitab Inc., State College, PA, USA). Hierarchical grouping analysis (HCA), considering the Euclidean distance and the complete linkage, was used to verify the similarity between the oil samples (OriginPro trial version, OriginLab Corporation, Northampton, MA, USA) [44].

## 4. Conclusions

The profiles of *Myrciaria* species showed significant chemical variability. This variability is related to different extraction methods, collection sites, plant parts, and genetic variability. Among the collected samples, five chemical profiles were reported for the first time: Profile I (*M. dubia*, α-pinene), Profile II (*M. floribunda*, terpinolene, α-phellandrene, and γ-terpinene), Profile III (*M. floribunda*, γ-cadinene, and an unidentified oxygenated sesquiterpene), Profile IV (*M. tenella*, *E*-caryophyllene and α-humulene), and Profile V (*M. tenella*, caryophyllene oxide and *E*-caryophyllene). It was impossible to infer changes in the significant constituents related to the influence of seasonality since none of the samples analyzed in the present work and in the literature aimed to monitor the variation in seasonal chemical composition. 

## Figures and Tables

**Figure 1 molecules-27-02234-f001:**
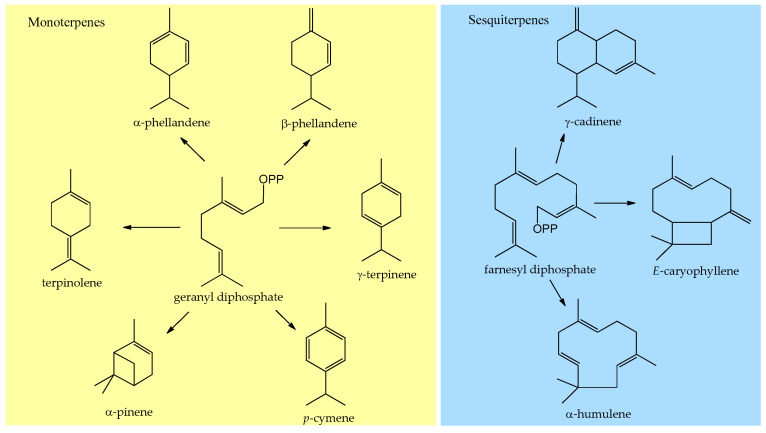
Primary mono- and sesquiterpenes arising from the geranyl and farnesyl diphosphate identified in the essential oils of *Myrciaria* species leaves. OPP = OPO_2_OPO_3_^−3^.

**Figure 2 molecules-27-02234-f002:**
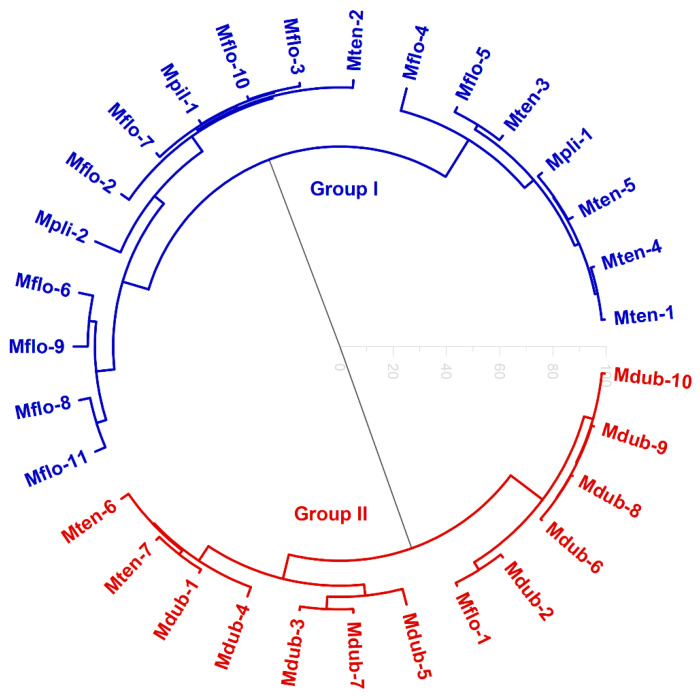
Hierarchical cluster analysis (HCA, in circular mode) of *Myrciaria* volatile samples.

**Figure 3 molecules-27-02234-f003:**
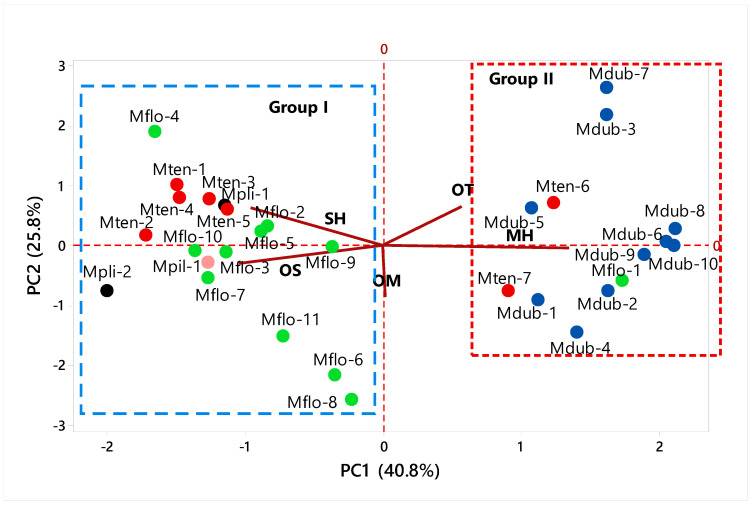
Principal component analysis (PCA) of *Myrciaria* volatile samples. *Myrciaria floribunda* (Mflo), *M. tenella* (Mten), *M. dubia* (Mdub), *M. plinioides* (Mpli), and *M. pilosa* (Mpil).

**Figure 4 molecules-27-02234-f004:**
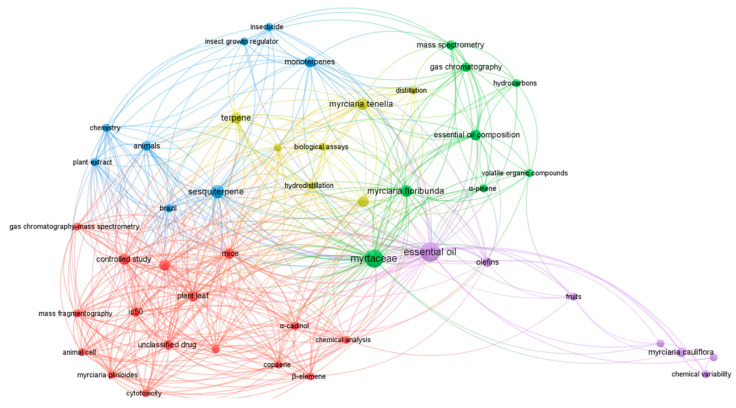
Network map of the most searched keywords and related to the theme, from 2010 to 2021.

**Table 1 molecules-27-02234-t001:** Yield and composition of essential oils from *Myrciaria* species leaves.

	RI_C_	RI_L_	Constituents (%)	*M. dubia*		*M. floribunda*		*M. tenella*	
				Mdub-1	Mdub-2	Mflo-1	Mflo-2	Mten-1	Mten-2
1	792	788 ^a^	2,4-dimethyl-3-pentanone	0.2					
2	845	846 ^a^	2*E*-hexenal			0.3			
3	847	850 ^a^	3*Z*-hexenol	0.1		0.3			
4	926	924 ^a^	α-thujene	0.1	0.6	3.7			
**5**	**936**	**932 ^a^**	**α-pinene**	**54.0**	**67.2**	**2.7**		**0.6**	
6	949	946 ^a^	camphene	0.2	0.1	0.1			
7	953	953 ^a^	thuja-2,4(10)-diene	0.2	0.1				
8	977	974 ^a^	β-pinene	0.8	1.5	0.2			
9	989	988 ^a^	myrcene			3.4			
10	1005	1008 ^a^	δ-3-carene		0.1	0.5			
**11**	**1006**	**1002 ^a^**	**α-phellandrene**			**17.7**			
12	1016	1014 ^a^	α-terpinene			3.3			
13	1024	1020 ^a^	*p*-cymene	0.5	1.3	7.2			
14	1028	1024 ^a^	limonene	3.8	3.7			0.1	
15	1029	1025 ^a^	β-phellandrene			6.6			
16	1031	1026 ^a^	1,8-cineole			1.4			0.1
17	1035	1032 ^a^	*Z*-β-ocimene			0.4			
18	1043	1044 ^a^	*E*-β-ocimene			0.3			
19	1057	1054 ^a^	γ-terpinene			8.7			
**20**	**1089**	**1086 ^a^**	**terpinolene**	**0.5**		**23.1**			
21	1098	1095 ^a^	linalool			0.4			
22	1099	1099 ^a^	α-pinene oxide	0.4	2.4				
23	1113	1114 ^a^	*endo*-fenchol	0.4					
24	1125	1122 ^a^	α-campholenal	1.5	0.7				
25	1138	1135 ^a^	*trans*-pinocarveol	2.9	0.6				
26	1144	1140 ^a^	*trans*-verbenol	0.1	2.3				
27	1161	1160 ^a^	pinocarvone	0.2	0.1				
28	1165	1165 ^a^	borneol	0.7					
28	1176	1174 ^a^	terpinen-4-ol	0.3		3.4			
30	1184	1179 ^a^	*p*-cymen-8-ol	0.5		0.8			
31	1190	1186 ^a^	α-terpineol	3.7	0.6	4.1		0.1	
32	1196	1194 ^a^	myrtenol	0.5	0.6				
33	1208	1204 ^a^	verbenone	0.3	0.6				
34	1218	1215 ^a^	*trans*-carveol	1.5	0.2				
35	1313	1316 ^a^	*Z*-patchenol		1.6				
36	1321	1325 ^a^	*p*-mentha-1,4-dien-7-ol		0.4				
37	1367	1367 ^a^	cyclosativene				1.6		
38	1371	1373 ^a^	α-ylangene				0.2		0.1
39	1375	1374 ^a^	α-copaene			0.1	0.1	0.3	1.2
40	1392	1389 ^a^	β-elemene				0.1	0.1	
41	1406	1400 ^a^	β-longipinene	0.2					
**42**	**1419**	**1417 ^a^**	***E*-caryophyllene**	**1.9**	**0.4**	**3.8**	**0.3**	**43.2**	**19.1**
43	1429	1430 ^a^	β-copaene			0.2		0.1	1.1
44	1438	1439 ^a^	aromadendrene		0.1			0.1	2.7
45	1443	1445 ^b^	selina-5,11-diene						0.2
46	1453	1452 ^a^	α-humulene	0.2		0.3	0.1	5.3	2.3
47	1456	1454 ^a^	*E*-β-farnesene					0.2	
48	1460	1464 ^a^	9-*epi*-*E*-caryophyllene						0.3
49	1461	1463 ^a^	*cis*-cadina-1(6),4-diene				0.2		
50	1475	1476 ^b^	selina-4,11-diene					0.6	
51	1476	1478 ^a^	γ-muurolene			0.1	0.1		1.7
52	1479	1483 ^a^	α-amorphene			0.1			0.2
53	1481	1480 ^a^	germacrene D				1.0		
54	1486	1489 ^a^	β-selinene			0.2	0.2	3.9	1.3
55	1495	1498 ^a^	α-selinene			0.2		2.8	1.1
56	1499	1500 ^a^	α-muurolene			0.2		0.1	0.6
57	1500	1499 ^a^	bicyclogermacrene				0.6		
58	1502	1508 ^a^	t*rans*-β-guaiene						
59	1508	1505 ^a^	β-bisabolene					2.2	
**60**	**1513**	**1513 ^a^**	**γ-cadinene**			**0.1**	**17.5**		**1.8**
61	1515	1514 ^a^	*Z-*γ-bisabolene					0.3	
62	1517	1520 ^a^	7-*epi*-α-selinene					0.2	
63	1522	1521 ^a^	*trans*-calamenene	0.1					1.5
64	1524	1513 ^a^	δ-cadinene			0.4	1.7		
65	1525	1528 ^a^	zonarene			0.1		0.2	
66	1531	1529 ^a^	*E*-γ-bisabolene					0.4	
67	1537	1537 ^a^	α-cadinene						0.3
68	1542	1544 ^a^	α-calacorene			0.1	0.9	0.3	
69	1545	1533 ^a^	flavesone	0.6					
70	1559	1559 ^a^	germacrene B				0.2		0.8
71	1561	1561 ^a^	*E*-nerolidol	0.2		0.1			
72	1564	1563 ^a^	β-calacorene				0.4		
73	1566	1566 ^a^	maaliol	0.1					0.1
74	1569	1570 ^a^	caryophyllenyl alcohol					1.2	0.2
75	1577	1577 ^a^	spathulenol	1.3	0.6		0.3		1.0
**76**	**1582**	**1582 ^a^**	**caryophyllene oxide**	**5.8**	**5.2**	**0.4**	**4.7**	**4.4**	**41.1**
77	1591	1592 ^a^	viridiflorol	0.3			0.3		0.4
78	1593	1595 ^a^	cubeban-11-ol	0.1				0.3	0.5
79	1599	1599 ^a^	longiborneol				0.7		
80	1600	1602 ^a^	guaiol				0.2		
81	1601	1600 ^a^	rosifoliol	0.2					0.9
82	1601	1602 ^a^	ledol					0.7	
83	1608	1608 ^a^	humulene epoxide II	0.5	0.3			0.4	2.4
84	1617	1621 ^a^	*iso*-leptospermone	1.5	0.1				
85	1618	1612 ^a^	1,10-di-*epi*-cubenol	1.1					
86	1626	1627 ^a^	1-*epi*-cubenol			0.1	1.8	0.9	2.3
87	1625	1629 ^a^	leptospermone	4.0	1.1				
88	1630	1628 ^a^	muurola-4,10(14)-dien-1β-ol				1.2		
89	1634	1639 ^a^	caryophylla-4(12),8(13)-dien-5β-ol	0.8				4.8	1.3
90	1635	1632 ^a^	*cis*-cadin-4-en-7-ol				0.4		
91	1638	1640 ^a^	hinesol					1.3	
92	1638	1638 ^a^	*epi*-α-cadinol						2.2
93	1640	1640 ^a^	*epi*-α-murrolol	0.2		0.1			
94	1644	1644 ^a^	α-muurolol			0.1	0.8	0.5	0.8
95	1646	1649 ^a^	β-eudesmol			0.1			0.3
96	1647	1639 ^a^	*allo*-aromadendrene epoxide					0.7	
97	1650	1652 ^a^	α-cadinol			0.2			1.6
98	1652	1658 ^a^	selin-11-en-4-α-ol					4.5	
99	1667	1656 ^a^	valerianol				2.2		
100	1663	1668 ^a^	*trans*-calamenen-10-ol						0.2
101	1663	1668 ^a^	14-hydroxy-9-*epi*-*E*-caryophyllene					2.9	
102	1665	1670 ^a^	*epi*-β-bisabolol					2.9	
103	1666	1661 ^a^	allohimachalol						0.7
104	1671	1675 ^a^	cadalene	0.1			4.3		0.4
105	1678	1683 ^a^	*epi*-α-bisabolol					0.3	
106	1680	1685 ^a^	germacra-4(15),5,10(14)-trien-1-α-ol				0.8	0.1	
107	1702	1701 ^a^	10-*nor*-calamenen-10-one				1.3		
**108**	**1745**		**Oxyg. sesquiterpene unidentified MW218 ^c^**				**15.0**		
109	1767	1772 ^a^	14-oxy-α-muurolene				1.7		
110	1779	1775 ^a^	guaiazulene				0.3		
111	1961	1959 ^a^	hexadecanoic acid					0.2	
Monoterpene hydrocarbons				60.3	74.6	77.9		0.7	
Oxygenated monoterpenoids				13.0	10.1	10.1		0.1	0.1
Sesquiterpene hydrocarbons				2.4	0.5	5.9	25.2	60.3	36.7
Oxygenated sesquiterpenoids				16.8	7.3	1.1	36.0	25.9	56.0
Others				0.1		0.6		0.2	
Total (%)				92.6	92.5	95.6	61.2	87.2	92.8
Oil yield (%)				0.5	1.1	1.4	1.5	2.1	1.2

**RI_C_** = calculated retention index using an n-alkane standard solution (C_8_–C_40_) in Rtx-5MS column; **RI_L_** = literature retention index. Main constituents in bold, n = 2 (standard deviation was less than 2.0); **Mflo** = *Myrciaria floribunda*; **Mten** = *M. tenella*; **Mdub** = *M. dubia*; **Mpli** = *M. plinioides*; **Mpil** = *M. pilosa*; ^a^ = Adams library [12]; ^b^ = FFNCS library [13]; ^c^ = Mass spectrum shown in Figure A1 (Appendix C).

**Table 2 molecules-27-02234-t002:** Collection site, herbarium voucher number, and geographic coordinates for the *Myrciaria* specimens.

Species	Code	Collection Site	Voucher Number	Coordinates Latitude/Longitude
*Myrciaria dubia*	Mdub-1	Belém, PA, Brazil	MG-229429	1°45′64.40″ S/48°43′86.75″ W
	Mdub-2	Castanhal, PA, Brazil	MG-063961	1°15′59.57″ S/48°01′7.66″ W
*Myrciaria floribunda*	Mflo-1	Belém, PA, Brazil	MG-228739	1°15′53.46″ S/48°8′11.52″ W
	Mflo-2	Belém, PA, Brazil	MG-229218	1°14′20.99″ S/48°26′10.24″ W
*Myrciaria tenella*	Mten-1	Baião, PA, Brazil	MG-237483	2°52′01″ S/49°29′08″ W
	Mten-2	Abaetetuba, PA, Brazil	MG-231854	1°45′15″ S/48°58′00″ W

## Data Availability

The data presented in this study are available on request from the corresponding author.

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
