# Peer review of "Chemical Composition and Variability of the Volatile Components of Myrciaria Species Growing in the Amazon Region"

_molecules, 2022, doi:10.3390/molecules27072234_

Round 1
Reviewer 1 Report
Based on the applications in the pharmaceutical, food, and cosmetic industries of Myrtaceae essential oils, the authors investigated the volatile chemical compositions for the species including M. dubia, M. floribunda, and M. tenella in the Brazilian Amazon. The chemotaxonomic study was further carried out based on the essential oil compositions from other Myrciaria species reported from Brazil and Colombia. This work is beneficial to meet the different needs of Myrtaceae essential oils obtained from different Myrtaceae species. These chemical compositions must be universally representative for every Myrtaceae specie to show the real chemical profiles. However, the chemical compositions in this paper obtained from different extraction methods, collection times, and plant parts, which will impair the validity of conclusions. Therefore, I cannot recommend this manuscript.
Author Response
Molecules-1628296, “Volatile Chemical Composition Variability of Myrciaria Species Growing in the Amazon Region”.
Responses to Reviewer 1
Question 1. Based on the applications in the pharmaceutical, food, and cosmetic industries of Myrtaceae essential oils, the authors investigated the volatile chemical compositions for the species including M. dubia, M. floribunda, and M. tenella in the Brazilian Amazon. The chemotaxonomic study was further carried out based on the essential oil compositions from other Myrciaria species reported from Brazil and Colombia. This work is beneficial to meet the different needs of Myrtaceae essential oils obtained from different Myrtaceae species. These chemical compositions must be universally representative for every Myrtaceae specie to show the real chemical profiles. However, the chemical compositions in this paper obtained from different extraction methods, collection times, and plant parts, which will impair the validity of conclusions. Therefore, I cannot recommend this manuscript.
Response: It seems to us that Reviewer 1's comment was confusing and unsubstantiated. The compositions of the oils of the Myrciaria species studied were all obtained and analyzed by the same methodology. It is evident that the comparison with literature data will always be differentiated depending on other distinct methodologies used. The chemotaxonomic study of the oils worked and conducted by us was well done and correctly. We are a group of researchers who have been publishing essential oil data for a few decades now. The comparison with samples already described in the literature could not be made differently. However, the results were fully compatible, regardless of the methodologies used. The reviewer was very sparing in his argument to criticize the results and the validity of the manuscript. See examples of previously published articles and their similarities:
doi.org/10.3390/plants10091854
doi.org/10.3390/ijms18122571
doi.org/10.1177/1934578X221080322
Prof. Dr. Pablo Luis. B. Figueiredo
UEPA, Belém, Brazil

Reviewer 2 Report
Comments and Suggestions for Authors
Title
It is recommended to revise the title and make it to be concise to the work.
Suggested title; Chemical composition and variability of the volatile components of Myrciaria Species Growing in the Amazon Region
Abstract
The sentence from line 26 to 30 is very lengthy and need to be rephrased.
Introduction
In general, the follow of information in introduction needs more improvement especially regarding the references and more details about the Myrciaria species. It was notices authors provide more details about Psidium, Eugenia and Syzygium genera despite the manuscript is about Myrciaria .
Results:
-Line70: essential oil yield (1.4-1.5%), followed by M. tenella (1.2-2.1%) and M. dubia (0.5-1.1%), please specify the exact yield for each sample of the six samples used separately
-Line 90: In table-1 please correct IR to RI and write % of constituents in columns of plant species and give the meanings of "Mdub-1, Mdub-2, Mflo-1, Mflo-2 Mten, and Mten” in table captions.
-Redesign the table 1 to add column for numbering the constituents and one more column for retention time.
-Provide the GC chromatograms either in the manuscript or as supplementary material
-Line 92 “c= Mass spectrum shown in appendix C” was mentioned but not found in table-1. Or appendix C itself , please revise
-Some components and values were bold please rational why these values and components are specified.
-Sections 2.1.1 to 2.1.3 are very confusing because it seems to be literature review and the results of current study are not clear and need a lot of improvements.
-Section2.1.4 Regarding chemical profiles for Myrciaria plinioides is not part of study so it is not a result. Similarly, section 2.1.5 about Myrciaria pilosa.
-Section 2.3. Biological activities of Myrciaria species Line 247 is also a literature not a result.
Materials and methods
Line 315: AR: awaiting registration at Herbarium of Emilio Goeldi Museum. I think it could not be accepted as not yet identified sample.
Line 318-319: it was mentioned that column used is Rtx-5MS (30 318 m × 0.25 mm; 0.25 μm film thickness) fused silica capillary column (Restek, Bellefonte, 319 USA) but in captions of table-1 line 91 it was mentioned “DB5-MS column”. Please specify?
Discussion part is missed
Conclusions
The authors stated that the variability may be due to the method of extraction and plant parts but in methodology, only one method was used “line 310- oil hydrodistillation using a Clevenger-type apparatus and line-305 collected plant part was leaves, So this conclusion is incorrect and need to be revised or the authors rational their opinion
For Appendix-B, please revise the abbreviations in table and meanings provided in captions
Author Response
Molecules-1628296, “Volatile Chemical Composition Variability of Myrciaria Species Growing in the Amazon Region”.
Responses to Reviewer 2
Title:
Question 1. It is recommended to revise the title and make it to be concise to the work.
Suggested title: Chemical Composition and Variability of the Volatile Components of Myrciaria Species Growing in the Amazon Region
Response: Suggestion accepted. It was revised.
Abstract:
Question 2. The sentence from line 26 to 30 is very lengthy and need to be rephrased.
Response: Suggestion accepted. It was revised.
Introduction:
Question 3. In general, the follow of information in introduction needs more improvement especially regarding the references and more details about the Myrciaria species. It was notices authors provide more details about Psidium, Eugenia and Syzygium genera despite the manuscript is about Myrciaria.
Response: Suggestion accepted. Is was revised. However, there are few studies with Myrciaria.
Results:
Question 4. Line70: essential oil yield (1.4-1.5%), followed by M. tenella (1.2-2.1%) and M. dubia (0.5-1.1%), please specify the exact yield for each sample of the six samples used separately
Response: Suggestion accepted. Is was revised.
Question 5. Line 90: In table-1 please correct IR to RI and write % of constituents in columns of plant species and give the meanings of "Mdub-1, Mdub-2, Mflo-1, Mflo-2 Mten, and Mten” in table captions.
Response. Suggestion accepted. Is was revised.
Question 6. Redesign the table 1 to add column for numbering the constituents and one more column for retention time.
Response: We've added a column with the compound numbers. However, a column with retention times will not bring more information to the table/manuscript. Today, retention time values are commonly transformed into retention index values, based on a homologous series of n-alkane hydrocarbons, eluted in the column under the same conditions. Thus, we could not attend to the reviewer's suggestion.
Question 7. Provide the GC chromatograms either in the manuscript or as supplementary material.
Response: OK, suggestion accepted. The GC chromatograms were provided in Figure A2, appendix C.
Question 8. Line 92: “c= Mass spectrum shown in appendix C” was mentioned but not found in table-1. Or appendix C itself, please revise.
Response: Please, see response to question 7.
Question 9. Some components and values were bold please rational why these values and components are specified.
Response: The bolded constituents are the main compounds. We input this information on the table footnote.
Question 10. Sections 2.1.1 to 2.1.3 are very confusing because it seems to be literature review and the results of current study are not clear and need a lot of improvements.
Response: Dear Reviewer, the manuscript aimed to provide information on the chemical composition variability of Myrciaria species to carry out the chemotaxonomic study. Statistical analyses were performed on the six studied specimens and those previously identified specimens in the literature. So, it was necessary to carry out a literature revision to compare all Myrciaria samples.
Question 11. Section 2.1.4 Regarding chemical profiles for Myrciaria plinioides is not part of study so it is not a result. Similarly, section 2.1.5 about Myrciaria pilosa.
Response: Please, see response to question 10.
Question 12. Section 2.3. Biological activities of Myrciaria species Line 247 are also a literature not a result.
Response: Please, see response to question 10.
Materials and methods:
Question 13. Line 315: AR: awaiting registration at Herbarium of Emilio Goeldi Museum. I think it could not be accepted as not yet identified sample.
Response: The plant samples have all been identified by the botanists at the Emílio Goeldi Museum and registered at the Murça Pires Herbarium. See Table 2 of the revised manuscript.
Question 14. Line 318-319: it was mentioned that column used is Rtx-5MS (30 318 m × 0.25 mm; 0.25 μm film thickness) fused silica capillary column (Restek, Bellefonte, 319 USA) but in captions of table-1 line 91 it was mentioned “DB5-MS column”. Please specify?
Response: The reviewer is correct. We used an Rtx-5MS column. Many thanks for your careful revision.
Conclusions
Question 15. The authors stated that the variability may be due to the method of extraction and plant parts but in methodology, only one method was used “line 310- oil hydrodistillation using a Clevenger-type apparatus and line-305 collected plant part was leaves, So this conclusion is incorrect and need to be revised or the authors rational their opinion.
Response: Dear reviewer, we compared our chemical results (obtained by hydrodistillation) with the research article found in the literature (obtained by different extraction methods), we described the methods used by authors. Thus, our conclusions agree with the results when compared with the findings in the literature.
Question 16. For Appendix-B, please revise the abbreviations in table and meanings provided in captions.
Response. OK, suggestion accepted. It was revised.
Prof. Dr. Pablo Luis. B. Figueiredo
UEPA, Belém, Brazil

Reviewer 3 Report
The article provides information of chemical composition variability of Myrciaria species in order to carry out chemotaxonomic study from the Amazon Region. The authors provided quantitative chemical profiling of essential oils from leaves of six Myrciaria specimens and performed statistical analysis from their specimens as well as from the literature available, previously identified specimens. The overall scientific soundness of the paper is solid; the English quality can be improved in several sentences that have a lots of commas.
My major concern is linked to the statistical analysis that combines literature data with the novel identified compounds, which is pretty useless when performed in this manner.
Major points
The authors use multivariate analysis to carry out chemotaxonomic study of different Myrciaria species, with the newly performed experiments from their laboratory along with the data from the literature. They also claim that Myrciaria chemical profiles shows significant variability in extraction methods, collection sites, plant parts, and genetic aspect. However, none of these problems is addressed before the multivariate analysis was performed leaving the results very questionable.
The results would be much more accurate if the statistical analysis were performed only on the data for newly preformed analysis from authors essential oils from the leaves of six Myrciaria species.
Minor points
Line 31: gas chromatography with mass spectrometer and flame ionisation detector
Lines 31&32: Add % (from-to) of named compound
Line 48: “producing which showing” please rephrase
Line 52: ”by their antifungal” rephrase to “because of their” or similar
Line 55: Microwave extraction does not give essential oils. Did you mean microwave distillation?
Line 72: GC-FID and GC-MS
Line 86: Figure 1. Explain the meaning of geranyl diphosphate and farnesyl diphosphate on the Figure 1.
Line 121: “specimens may be related”. Different extraction methods, plant parts and/or collection sites by all means result in different chemical compounds, there is no “may be”.
Line 178: Were all the specimen samples taken at the same time? If not, the sampling period can be addition factor for differences in chemical profiles.
Line 190: Reference for genetic factor contribution?
Line 247: Paragraph “2.3. Biological activities of Myrciaria species” does not belong in Result since you did not perform any of listed experiments
Line 277: The explanation for bibliometric network map (first sentence) is unclear, please rephrase.
Line 305: Were the leaves collected at particular point in certain month, or the leaves were collected during several months and jointly distilled?
Line 312: “ml/100g”? How did you determine volume of the oil?
Line 331: In how many repetitions have you analysed the oils? Please add mean values into Table 1.
Line 363: “this variability may be related”. Different extraction methods, collection sites, plant parts and genetic variability by all means result in different chemical compounds, there is no “may be”.
Line 422: Have you subtracted the background from the sample? Or change the temperature program to see if this peak is a mixture of two compounds?
Author Response
Molecules-1628296, “Volatile Chemical Composition Variability of Myrciaria Species Growing
in the Amazon Region”.
Responses to Reviewer 3
The article provides information of chemical composition variability of Myrciaria species in order to carry out chemotaxonomic study from the Amazon Region. The authors provided quantitative chemical profiling of essential oils from leaves of six Myrciaria specimens and performed statistical analysis from their specimens as well as from the literature available, previously identified specimens. The overall scientific soundness of the paper is solid; the English quality can be improved in several sentences that have a lot of commas.
My major concern is linked to the statistical analysis that combines literature data with the novel identified compounds, which is pretty useless when performed in this manner.
Major points
Question 1. The authors use multivariate analysis to carry out chemotaxonomic study of different Myrciaria species, with the newly performed experiments from their laboratory along with the data from the literature. They also claim that Myrciaria chemical profiles shows significant variability in extraction methods, collection sites, plant parts, and genetic aspect. However, none of these problems is addressed before the multivariate analysis was performed leaving the results very questionable.
Response: Dear reviewer, one of our objectives was to compare the different profiles of Myrciaria volatiles described in the literature with the composition of the six essential oils of Myrciaria da Amazônia, extracted by hydrodistillation. Every time the literature reported a different extraction method, this method was described (sections 2.1.1 to 2.1.5) and appeared in Table A1. Our intention was not to omit any factors that could differentiate samples of the same species.
Question 2. The results would be much more accurate if the statistical analysis were performed only on the data for newly preformed analysis from authors essential oils from the leaves of six Myrciaria species.
Response: Yes, indeed, the results would be more accurate if the statistical analysis were performed only with the oils from the six Myrciaria specimens we reported. However, the objective of the work was to carry out a complete chemotaxonomic study, comparing the results with the samples described in the literature. The methodology used is following several articles published in the literature:
doi.org/10.3390/plants10091854;
doi.org/10.3390/ijms18122571
doi.org/10.1177/1934578X221080322
Minor points
Question 3. Line 31: gas chromatography with mass spectrometer and flame ionization detector.
Response: The text has been revised.
Question 4. Lines 31&32: Add % (from-to) of named compound.
Response: The text has been revised.
Question 5. Line 48: “producing which showing” please rephrase.
Response: The text has been revised.
Question 6. Line 52: ”by their antifungal” rephrase to “because of their” or similar.
Response: The text has been revised.
Question 7. Line 55: Microwave extraction does not give essential oils. Did you mean microwave distillation?
Response: The text has been revised.
Question 8. Line 72: GC-FID and GC-MS.
Response: The text has been revised.
Question 9. Line 86: Figure 1. Explain the meaning of geranyl diphosphate and farnesyl diphosphate on the Figure 1.
Response: Geranyl diphosphate and farnesyl diphosphate are the precursors of mono and sesquiterpenes in the secondary plant metabolism.
Question 10. Line 121: “specimens may be related”. Different extraction methods, plant parts and/or collection sites by all means result in different chemical compounds, there is no “may be”.
Response: The text has been revised.
Question 11. Line 178: Were all the specimen samples taken at the same time? If not, the sampling period can be addition factor for differences in chemical profiles.
Response: The plants worked by us were all collected at the same time. Concerning plant data reported in the literature, this information was not accurate.
Question 12. Line 190: Reference for genetic factor contribution?
Response: The text has been revised.
Question 13. Line 247: Paragraph “2.3. Biological activities of Myrciaria species” does not belong in Result since you did not perform any of listed experiments.
Response: Item 2 of manuscript has been revised to "Results and Discussion". Item 2.3 should be considered an item for discussion of the results in comparing Myrciaria oils worked by us with those previously reported in the literature.
Question 14. Line 277: The explanation for bibliometric network map (first sentence) is unclear, please rephrase.
Response: The text has been revised.
Question 15. Line 305: Were the leaves collected at particular point in certain month, or the leaves were collected during several months and jointly distilled?
Response: The leaves were collected at a single point for several months. The process of obtaining the oils was individual for each field collection, month-by-month.
Question 16. Line 312: “ml/100g”? How did you determine volume of the oil?
Response: The oil volume was measured in a centrifuge tube after dehydration and centrifugation.
Question 17. Line 331: In how many repetitions have you analysed the oils? Please add mean values into Table 1.
Response: The text and table 1 has been revised. See the table 1 footnote.
Question 18. Line 363: “this variability may be related”. Different extraction methods, collection sites, plant parts and genetic variability by all means result in different chemical compounds, there is no “may be”.
Response: The text has been revised.
Question 19. Line 422: Have you subtracted the background from the sample? Or change the temperature program to see if this peak is a mixture of two compounds?
Response: When analyzing the chromatogram of total ions and the mass spectrum, good selectivity and resolution were observed for this compound. It is not a mixture of isomers. Probably a newly oxygenated sesquiterpene. We will try to separate it for identification by NMR in the future.
Prof. Dr. Pablo Luis. B. Figueiredo
UEPA, Belém, Brazil

Round 2
Reviewer 1 Report
The revised manuscript is OK. I recommend this revised version to publish.
Author Response
We are very grateful for the comments of the reviewer 1.
Reviewer 2 Report
The authors answered most of comments to improve the quality of manuscript‎.Author Response
We are very grateful for the comments of the reviewer 2.
Reviewer 3 Report
Major point:
If the authors insist on performing statistical analysis from their experiments as well as data from the literature- ok, but use the data for the essential oils only. Or SPME only. Do not mix results from different extraction techniques. The result of volatiles from the essential oil and by SPME of the same plant part sampled in the same day at the same location result in substantial different chemical profile, adding additional confusion for the interpretation of the results.
Minor points:
Line 86: Figure 1. Explain the meaning of geranyl diphosphate and farnesyl diphosphate on the Figure 1.
Thank you for the clarification, but you still haven't made any change in the desription line of Figure 1 and/or in the main text. Please explain there as well.
Line 344. “The analyses were triplicate“ Please add „made in triplicate“ or similar
Author Response
Major point:
If the authors insist on performing statistical analysis from their experiments as well as data from the literature- ok, but use the data for the essential oils only. Or SPME only. Do not mix results from different extraction techniques. The result of volatiles from the essential oil and by SPME of the same plant part sampled in the same day at the same location result in substantial different chemical profile, adding additional confusion for the interpretation of the results.
Response: Dear Reviewer, one of the manuscript's objectives is to describe the different chemical volatile profiles (not essential oils) of Myrciaria that can be obtained by various collection sites and extraction methods. The multivariate analysis that we performed gives us this information. Furthermore, all samples were analyzed by GCMS, which allows us to perform PCA and HCA analyses. Still, similar analyzes can be found in the articles published in the MDPI as follows.
doi.org/10.3390/plants10091854
doi.org/10.3390/ijms18122571
doi.org/doi.org/10.1177/1934578X221080322
Finally, unfortunately, we are unable to comply with this suggestion
We believe the comparisons to be both valid and informative.
Minor points:
Line 86: Figure 1. Explain the meaning of geranyl diphosphate and farnesyl diphosphate on the Figure 1.
Thank you for the clarification, but you still haven't made any change in the desription line of Figure 1 and/or in the main text. Please explain there as well.
Response: Corrected, we changed the figure description to “Primary mono- and sesquiterpenes arising from the geranyl and farnesyl diphosphate identified in the essential oils of Myrciaria species leaves. OPP = OPO2OPO3-3 “.
Line 344. “The analyses were triplicate“. Please add „made in triplicate“ or similar
Response: Corrected.